# Diaryl-Pyrano-Chromenes Atropisomers: Stereodynamics and Conformational Studies

**DOI:** 10.3390/molecules28134915

**Published:** 2023-06-22

**Authors:** Alessia Ciogli, Andrea Fochetti, Andrea Sorato, Giancarlo Fabrizi, Nunzio Matera, Andrea Mazzanti, Michele Mancinelli

**Affiliations:** 1Department of Chemistry and Drug Technologies, Sapienza University of Rome, Piazzale Aldo Moro 5, 00185 Roma, Italy; andrea.fochetti@uniroma1.it (A.F.); andrea.sorato@uniroma1.it (A.S.); giancarlo.fabrizi@uniroma1.it (G.F.); 2Department of Industrial Chemistry “Toso Montanari”, University of Bologna, Viale del Risorgimento 4, 40136 Bologna, Italy; nunzio.matera2@unibo.it (N.M.); andrea.mazzanti@unibo.it (A.M.)

**Keywords:** aryls-pyrano-chromenes, atropisomer, Dynamic-NMR, Dynamic-HPLC, ECD, DFT, TD-DFT

## Abstract

The dynamic scenario of di-aryls-pyrano-chromenes was investigated using DFT calculations. The symmetry of the chromene scaffold and the presence of two *ortho*-substituted aryls substituents can generate two *syn*/*anti* diastereoisomers and conformational enantiomers with different rotational barriers. The relative conformations and configurations were derived using NOESY-1D experiments. Depending on the energies related to the conformational exchange, the experimental energy barriers were determined through Dynamic NMR, Dynamic HPLC or kinetic studies. The atropisomeric pairs were resolved in the latter scenario, and their absolute configuration was assigned using the ECD/TD-DFT method.

## 1. Introduction

In recent years, atropisomeric compounds have received significant attention due to their numerous applications, and the synthesis of new entities is a rapidly growing research area [1,2,3]. They are often involved in pharmaceuticals, are present in bioactive natural products [4,5,6] and are frequently used as chiral ligands in asymmetric synthesis [7,8,9,10,11,12,13]. The most well-known and perhaps explored atropisomers are biaryl systems, spiranes and compounds with a C_sp2_-N [14,15,16,17] or C_sp2_-B [18,19] axis bearing at least one chiral axis. Interestingly, when two stereogenic axes and sufficiently hindered substituents around the chiral axis are present, molecules with a potential “cleft” structure could be prepared [20,21,22,23,24]. In these cases, a deeper investigation concerning the rotational energy barriers of each axis could provide valuable support in developing rigid atropisomeric clefts with potential applications in molecular recognition, such as chiral catalysts or molecular machines [25,26].

Recently, Fochetti et al. reported the synthesis of 4,6-diphenyl-2*H*,8*H*-pyrano [3,2-g]chromene (hereafter referred to as 4,6-diaryls-pyrano-chromene **1**) and 4,10-diphenyl-2*H*,8*H*-pyrano[2,3-f]chromene (hereafter referred to as 4,10-diaryls-pyrano-chromene **2**) through a gold (I)-catalyzed intramolecular hydroarylation reaction (IMHA), which is compatible with different functional groups on the aryls [27]. In this paper, we reported the possibility of introducing two chiral axes by increasing the steric hindrance of the substituent in the ortho positions of the aryls (Figure 1). The aryls are indeed skewed with respect to the dynamically planar scaffold of chromene, and they could be driven to be almost perpendicular to it by raising the steric hindrance in the ortho position. In this conformation, we formed two stereogenic axes that, in principle, generate two diastereoisomers (*syn*/*anti*) with four conformations for the 4,10-diaryls-pyrano-chromene derivatives **2** and three conformations in the case of 4,6-diaryls-pyrano-chromene derivative **1**, due to the *C*_s_ symmetry of the *syn* stereoisomer.

The resulting diastereomeric conformations can be either stereolabile or configurationally stable. Herein, we explored the dynamic scenario of di-aryls-pyrano-chromenes using DFT calculations, and depending on the *syn*/*anti* conformational exchange, we were able to determine the experimental free energy barriers through Dynamic NMR, Dynamic HPLC or Kinetic studies. Moreover, when the interconversion barriers were high enough, we isolated the atropisomeric compounds and assigned their absolute configuration using the ECD/TD-DFT method.

## 2. Results and Discussion

### 2.1. Synthesis

The synthesis of compounds **1a** and **2a** was obtained, as reported by some of the authors [27]. 1,3-bis((3-bromo-prop-2-yn-1-yl)oxy)benzene was achieved from resorcinol and propargyl bromide, followed by Sonogashira’s cross-coupling with 2-bromoiodobenzene. Intramolecular hydroarylation in the presence of a gold catalyst yielded the two regioisomers (“linear” **1a** and “bent” **2a**) in a 70:30 ratio. For compounds **1b**/**2b**, the developed protocol required a slight modification, introducing the triple bond into the aryl moiety (in this case, 2-methyl naphthalene), followed by the reaction with resorcinol, as reported in Figure 2. 1-(3-bromoprop-1-yn-1-yl)-2-methylnaphthalene **4** was obtained in four steps using 1-bromo-2-methylnaphthalene **3** as the starting material. Subsequent alkylation of resorcinol, performed with K_2_CO_3_ in DMF at room temperature, gave the intermediate 1,3-bis((3-(2-methylnaphthalen-1-yl)prop-2-yn-1-yl)oxy)benzene, **5 [27,28]**. The intramolecular hydroarylation (IMHA) of compound **5** in the presence of a gold catalyst yielded compounds **1b** and **2b** with an 87/13 ratio (Figure 2).

### 2.2. DFT Calculations

The investigation of atropisomer stability was carried out by DFT calculations, and the results were supported by HPLC and/or NMR at variable-temperature (dynamic-HPLC and dynamic-NMR). As a general remark, the DFT calculations assume that two diastereoisomers for each compound, syn/anti, are generated by the restricted C_sp2_-aryl rotation. In terms of stereoisomers, compound **1** yields only three stereoisomers due to the two anti-enantiomeric conformations (C_2_ point group) and the meso/syn conformation (C_s_ point group), while compounds **2**, with an asymmetric pyrano-chromene scaffold, yield four stereoisomers (as shown in Figure 3).

The DFT study began with the ortho-bromide phenyl substituted pyrano-chromenes (**1a** and **2a**), then moved on to the more sterically hindered compounds (**1b** and **2b**). As in similar systems, the activation entropy was determined to be negligible [23,24], and the ZPE-corrected enthalpy was considered in the calculation of rotational barriers and was consistently in agreement with the experimental barriers. The free energy term as is, or after frequency cut-off at 100 cm*^−^*^1^ (Appendix A) [29] and also including empirical dispersion as B3LYP-D3 (Appendix A), did not fit with the experimental data. All the ground states (GS) and transition states (TS) geometries for compounds **1b**, **2a** and **2b** are reported in the Supporting Information. The syn/anti calculated conformations of compound **1a** at the B3LYP/6-311++G(d,p) level [30], including chloroform as solvent (IEF-PCM approach) [31,32], are displayed in Figure 1, and Table 1 reports all the descriptors identifying the different conformations.

Six conformations were found for compound **1a**, depending on the flexibility of the pyrano-chromene scaffold. Looking towards it, the oxygens could be considered in the plane, while the CH_2_ of pyrano-chromene are out of the plane and could be either towards or away from the bromine atoms, generating three conformations for both *anti* and *syn* diastereoisomers. The *syn*/*anti* interconversion occurs through the transition state TS1_ext_ where the aryl group rotates with the bromine outside the pyrano-chromene plane. The calculated enthalpy energy barrier was 66.5 kJ/mol (15.9 kcal/mol) for compound **1a**.

Similar conformations were found in compound **1b** (Appendix A), where the steric hindrance of the 2-methyl-naphthyl group raised the energy barrier to 130.5 kJ/mol (31.2 kcal/mol). The TS2_EXT_ shows a highly distorted conformation where the *ortho*-methyl is outside the plane while the naphthyl ring is inside. This high barrier implies that the two *syn*/*anti* configurations could be isolated at room temperature.

For compounds **2a** and **2b**, the corresponding *anti-to-syn* interconversion barrier is due to the rotation of the aryl ring with the lower barrier (Appendix A). In contrast, the higher energy barrier leads to racemization. To distinguish the two aryl groups, in Figure 3 and Appendix A, the pyrano-chromene scaffold and the corresponding bonded aryl group were labelled as A and B, respectively. The asymmetry of the scaffold generated four transition states, of which only two are effective. Focusing on compound **2a**, the aryls could rotate with the bromine inside the plane (named INT) or outside the plane (named EXT), while the other aryl group does not turn (named FIX). In detail, the transition state with the lowest energy barrier was found to be TS2_A-EXT-B-FIX_ (65.3 kJ/mol or 15.6 kcal/mol). This barrier interconverts the *syn* into *anti*-diastereoisomers (hereafter referred to as the diastereomerization barrier). The second aryl group can rotate with a calculated energy of 70.7 kJ/mol (16.9 kcal/mol, TS3_A-FIX-B-EXT_). This barrier interconverts the enantiomers (hereafter referred to as the enantiomerization barrier).

Compared to compound **1b**, compound **2b** has diastereomerization and enantiomerization barriers higher than 125.5 kJ/mol (30 kcal/mol). Therefore, their separation should be easily achieved by chromatographic techniques.

A comprehensive comparison of calculated vs experimental data (population of conformers and rotational barriers) can be found at the bottom of the text (Table 2 and Table 3).

### 2.3. Dynamic NMR

Due to the low energy calculated for the diastereomerization process (TS1_ext_ = 66.5 kJ/mol for **1a** and TS2_A-EXT-B-FIX_ = 65.3 kJ/mol for **2a**), variable-temperature NMR (D-NMR) is the most appropriate technique to confirm these predicted values experimentally. The flexibility of the oxygen bridges in the chromene moieties makes the ring inversion very fast and averaged over the NMR time scale. The chromene scaffold can be dynamically planar at any temperature reachable by the D-NMR technique. After HPLC separation of the **1a**/**2a** mixture, the linear **1a** and bent **2a** were subjected to variable-temperature ^1^H NMR investigation.

For compound **1a**, the alkenyl ^1^H NMR region showed broadened signals at +26 °C (Figure 2). The ^1^H NMR signal of the aromatic CH between the two oxygens (H-10) splits into two singlets at −12 °C. Similarly, the alkenyl signal splits into two triplets. At this temperature, the aryl-chromene rotational barrier is frozen in the NMR time scale, showing anisochronous signals corresponding to the two syn/anti diastereoisomers.

Albeit, for different reasons, the two methylene hydrogens for diastereoisomers have different magnetic environments and are diastereomeric. This means they do not help detect which signals belong to the chiral anti-diastereoisomer. However, the assignment of the anti-chiral conformation was achieved by acquiring the ^1^H NMR spectrum in a chiral environment (Figure 3), by adding (*R*)-2,2,2-trifluoro-1-(9-anthryl)-ethanol (R-TFAE) [33] to a solution of compound **1a** in CD_2_Cl_2_ at −20 °C (18.2:1 molar ratio R-TFAE:**1a**). Only the signal of H-10 of the anti-chiral isomer splits into a pair of diastereotopic singlets, corresponding to the M,M and P,P enantiomers, while the same hydrogen for the C_s_-symmetric syn conformation does not further split. In this way, it was possible to determine the syn/anti populations as 44.2:55.8.

When the temperature is raised, the alkenyl triplet and the single aromatic signals exhibit line broadening due to the syn/anti exchange. Above +82 °C, when the interconversion process becomes fast, these signals become narrow and well-defined triplet and singlet signals, respectively. By line shape simulations of the spectra at different temperatures and using the Eyring equation, a ΔG^≠^ value of 64.0 kJ/mol (15.3 kcal/mol) is obtained, corresponding to the barrier for the interconversion of the anti into the syn conformer. Due to the symmetrical scaffold of **1a,** the syn/anti exchange can be achieved by rotating a single aryl ring. Thus, to derive the barrier corresponding to the rotation of a single aryl ring, the values of the rate constants reported must be halved, yielding a barrier of 65.7 kJ/mol (15.7 kcal/mol) [23,24]. This value is in perfect agreement with the DFT calculated enthalpy energies (see Figure 1).

The D-NMR spectra for compound **2a** showed a similar trend (see Appendix A). The vinylic ^1^H NMR signal at 5.77 ppm in C_2_D_2_Cl_4_ at each temperature was simplified from ABX to AB system by homodecoupling of the CH_2_. In this way, at +25 °C, two doublets, at 4.83 ppm and 4.70 ppm with a ^2^J_AB_ = 14 Hz, were visible, and they split into two AB systems at −10 °C, indicating the presence of both *anti* and *syn* conformations. The experimental diastereomerization *syn/anti*-energy barrier has been found by DNMR simulations to be 65.3 kJ/mol (15.6 kcal/mol). However, in the case of **2a**, both diastereoisomers are chiral, and it was not experimentally possible to assign the *anti*:*syn* ratio. DFT calculations suggest that the **2a**-*syn* has a larger dipole moment with respect to **2a**-*anti*. Therefore, the *syn/anti* ratio was evaluated in solvents with increasing polarity (CDCl_3_ vs CD_3_CN). The calculated syn/anti ratio of 39/61 (PCM: chloroform) agrees with the experimental ^1^HNMR ratio 38.5/64.5 recorded in CDCl_3_ at −10 °C. Again, the calculated syn/anti ratio of 44/56 (PCM: acetonitrile) agrees with the experimental ^1^HNMR ratio of 43.4/56.6 recorded in CD_3_CN. As expected, the syn population slightly increases in more polar solvents (Appendix A). Both spectra were recorded at −10 °C to have a better resolution for the signals since the diastereomerization barrier is frozen.

On raising the temperature above +25 °C, the AB system coalesced around +90 °C and became a singlet at +114 °C (Appendix A). This dynamic phenomenon is due to the more hindered B aryl group that exchanges the two conformational enantiomers still present when the syn/anti exchange is fast. Line shape simulation of the ^1^H NMR spectra allowed the determination of the enantiomerization energy barrier as 74.0 kJ/mol (17.7 kcal/mol).

### 2.4. Dynamic HPLC

Diastereomerization and enantiomerization processes for compounds **1a** and **2a** could, in principle, be detected using Dynamic CSP-HPLC at lower temperatures with respect to NMR. For compound **1a**, we attempted dynamic-CSP-HPLC at low temperatures, down to −63 °C, but the amylose-based stationary phase did not provide valuable data. On the other hand, four broad peaks were observed for compound **2a** at −63 °C, confirming the presence of four stereoisomers (Figure 4). However, the online chiro-optical detection at −63 °C was unfeasible, and the fast rotation at room temperature prevented the assignment of each enantiomer.

As the temperature increased, the lowest barrier was exceeded, and the A-aryl group was free to rotate, separating two broad peaks at −48 °C. With further temperature increase, a chromatographic trace recorded at −36 °C showed a typical plateau between two peaks. When the B-aryl group was free to rotate, a single peak was detectable over +3 °C. The energy of the enantiomerization barrier (TS3_A-FIX-B-EXT_ of Appendix A) was estimated to be 70.7 kJ/mol (16.90 kcal/mol) through simulation of the chromatogram trace [33,34,35,36]. Also, in this case, the calculated enthalpy energy barrier was in perfect agreement with the experimental data.

### 2.5. Separation and Characterization of Atropisomeric Compounds **1b** and **2b**

In the case of compounds **1b** and **2b**, the conformational features are similar to their corresponding **1a** and **2a**, respectively. However, the 2-Methylnaphthyl substituent significantly increases the rotational energy barrier [37,38,39]. The calculated energy barrier is higher than 30 kcal/mol, which is high enough to allow the separation of the syn/anti isomers and atropisomers resolution at room temperature.

For compound **1b**, we were able to separate the syn/anti diasteroisomers using CSP-HPLC on (R,R)-Whelk-O1 5 μm column (150 × 4.6 mm L × ID) with n-hexane/dichloromethane 95/5 as eluent. The anti and syn diastereoisomers were assigned based on the opposite CD signals for the two anti-enantiomers, while the peak of the syn stereoisomers did not show any CD signal (Figure 5).

Compound **2b** was resolved into the four available stereoisomers using CSP-HPLC on a Chiralpak IB-N5 5 μm column (250 × 10 mm L × ID) with n-hexane/chloroform 85/15 as eluent (Appendix A). After separation, ^1^H-NMR spectra of the four stereoisomers were acquired, and the syn/anti-assignment was unambiguously achieved using the double pulsed field gradient spin-echo NOE (DPFGSE-NOE) sequence [40,41,42,43]. In this case, signals corresponding to the ortho-methyl groups are the most useful for the NOE analysis. In the anti-configuration, the methyl groups are too far away to yield any NOE enhancement. For the second and the third eluted stereoisomers, besides having identical ^1^H NMR spectra, the signals, at 2.16 ppm and 2.38 ppm, respectively, did not show any NOE effect, attesting to their enantiomeric relationship of the anti-isomer. However, the first and fourth peaks obtained through CSP-HPLC separation showed an NOE effect elucidating that the methyl groups are on the same side as for the syn atropisomer. Specifically, when the methyl at 2.44 ppm for the syn atropisomer was irradiated, a slight NOE effect on the second methyl signal at 2.30 ppm was detected (Figure 6).

### 2.6. Kinetic Studies for Compounds **1b** and **2b**

Being very high in the energy barrier involved, rate constants of the syn/anti interconversion for compounds **1b** and **2b** were derived from kinetic studies using either HPLC or ^1^H NMR as the monitoring techniques. In detail, for the compound **1b**, syn/anti-stereo-stability was investigated starting from the pure syn **1b** in cis/trans-decaline solution and monitoring the appearance of anti **1b** at different temperatures (+110 °C, +120 °C and +130 °C respectively, see Appendix A and details in SI). The diastereomerization processes were completed in almost 4 h, providing energy barriers of about 129.3 kJ/mol (30.9 kcal/mol), matching the prediction from DFT calculations. The ΔG^#^ value results were slightly affected by the temperature attesting a low contribution of entropic term and confirmed by Eyring plot analysis, where the extrapolated ΔS was −12 u.e.

For compound **2b**, the monitoring over the time of the ^1^H NMR methyl signal of the pure anti-isomer in a C_2_D_2_Cl_4_ solution kept at different temperatures allowed valid results (Appendix A). The methyl ^1^H NMR signal of anti-decreases while that one of the syn began to appear. After 48 h at 120 °C the equilibrium was reached with a syn/anti ratio of 50:50. Using a first-order kinetics equation for a process at equilibrium, we obtained the rate constants at different temperatures, hence the energy barrier (133.5 kJ/mol, 31.9 kcal/mol).

### 2.7. Assignment of Absolute Configuration for Compounds **1b** and **2b**

Compounds **1b** and **2b** do not contain heavy atoms in their structures, making it impossible to assign their absolute configuration (AC) using anomalous dispersion X-ray diffraction with Mo-Kα radiation. Instead, we used chiroptical properties, namely electronic circular dichroism spectroscopy (ECD), in synergy with time-dependent density functional theory (TD-DFT) [44]. For compound anti-**1b**, the ECD spectra recorded in acetonitrile of the first eluted CSP-HPLC (Figure 7) showed a small positive band at 280 nm, a large negative cotton effect at 235 nm and a large positive band at 215 nm due to the interactions between the dipoles of pyrano-chromene and the 2-methyl-naphthyl moieties (Appendix A). Being the three ground states for the anti-conformation very close in energy, the ECD spectrum for each ground state was calculated (Appendix A), and the weighted average was obtained considering the Boltzmann population (Figure 7). ECD simulations were obtained using four different functionals for redundancy and the 6-311++G(2d,p) basis set with acetonitrile as solvent (IEF-PCM approach) [31,32]. The calculated ECD for ***P,P*** showed a good agreement with the experimental spectrum of the second eluted CSP-HPLC; consequently, we assigned the ***M,M*** AC to the first eluted atropisomer anti-**1b**.

The relative configurations of compound **2b** were assigned by the NOE effect (see Figure 5); subsequently, the AC was assigned in the same way as compound **1b**. The details were reported in the Appendix A. The *syn P_A_,P_B_* AC was assigned to the first eluted, the *syn M_A_,M_B_* AC to the fourth, while the *anti P_A_,M_B_* AC was assigned to the second eluted and the *anti M_A_,P_B_* AC to the third (Appendix A for Mos involved in the UV transitions).

## 3. Materials and Methods

### 3.1. Materials

All reagents and solvents, HPLC and ACS grade, were purchased from Sigma Aldrich (Milan, Italy). HPLC gradient grade solvents were filtered on 0.45 μm Omnipore filters (Merck Millipore, Darmstadt, Germany) before use. Analytical-grade solvents and commercially available reagents were used as received. Deuterated solvents (CDCl_3_, C_2_D_2_Cl_4_, DMSO-d6) for NMR spectra were obtained from EurisoTop. The following stationary phases were employed for the chromatography: silica gel 60 Å F254 (Merck) for TLC and silica gel 60 Å (230−400 mesh, Sigma-Aldrich) for atmospheric pressure chromatography. The glassware used in these reactions was placed in an oven at 70 °C for at least 3 h immediately before use.

### 3.2. Semipreparative HPLC

A Waters 600 HPLC pump, a Rheodyne7012 injector (loop of 1 mL) and a Waters 2487 UV detector (Waters, Milford, MA, USA) with a wavelength set at 254 nm were used to purify the products at semi-preparative level. The stationary phases employed are reported in the product characterizations.

### 3.3. HPLC System and Simulation of Dynamic Profile

HPLC analyses were performed on the Jasco HPLC system equipped with a PU-980 HPLC pump, a Rheodyne 7725i injector (loop of 20 µL), a 975 series UV detector and a 995-CD series detector. Chromatographic data were recorded and processed with Borwin software (Version 1.50, Jasco Europe, Italy). Variable low-temperature HPLC chromatograms were obtained on a Jasco HPLC by submerging the column in a dewar containing dry ice and acetone. The uncertainty in temperature measurements can be estimated as ±0.5 °C. Dynamic HPLC plots were simulated with the Auto-D-HPLC-Y2 K software based on a stochastic model. This software performs a line-shape simulation by the simplex algorithm to optimize chromatographic and kinetic parameters, thus obtaining the best agreement between the experimental and simulated dynamic profiles. The error in determining the activation-free energies can be estimated as ±0.2 kcal/mol.

### 3.4. HPLC Analysis

Sample **1a**: unresolved at −63 °C.

Sample **2a** was analyzed on Chiralpak-IA (250 × 4.6 mm, L × I.D.) with eluent *n*-hexane-dichloromethane 98/2 + 0.5% ethanol at 1.0 mL/min and 254 nm UV detection (sharp peak at 8.1 min and 25 °C). Chromatographic traces recorded at 25 °C, 3 °C, −36 °C, −48 °C and −63 °C.

Sample **1b** was investigated on (*R*,*R*)-Whelk-O1 5µm column (150 × 4.6 mm, L × I.D.) with n-hexane-dichloromethane 95/5 at the flow rate of 1.0 mL/min, detectors UV at 254 nm and CD at 280 nm (retention time of 1b-(−)-anti 13.6 min, 1b-(+)-anti 14.8 min and 1b-syn 17.8 min). Semipreparative conditions: (*S*,*S*)-Whelk-O2 10µm (250 × 10 mm) with *n*-hexane-dichloromethane 95/5 at a 4.0 mL/min flow rate and 254 nm UV detection.

Sample **2b** was analyzed on Chiralpak IB-N5 5 µm column (250 × 10 mm L × ID) with *n*-hexane/chloroform 85/15 as eluent.

### 3.5. Off-Line Kinetic Procedure

Compound **1b**

An aliquot of a pure atropisomer was dissolved in 1 mL of cis/trans decalin (boiling point = +189 °C – +191 °C) using a test tube with a screw cap. Then the sample was heated in a thermostatic oil bath (±2 °C) at the desired temperature. Small samples were taken at different times directly in a syringe containing the mobile phase. The enantioselective HPLC then allowed the determination of the enantiomeric ratio.

Compound **2b**

An aliquot of syn atropisomer was dissolved in 0.7 mL of C_2_D_2_Cl_4_ (boiling point = +146 °C) using an NMR tube. Then the sample was heated in a thermostatic oil bath (±2 °C) at the desired temperature. After cooling fast at ambient temperature, the ^1^H NMR was acquired at different times. The anti-atropisomer formation was monitored, and the integral of methyls was taken to calculate the ratio syn/anti. A first-order kinetic equation was then used to derive the rate constant for diastereomerization and, hence, the activation barrier using the Eyring equation.

### 3.6. NMR Experiments

NMR spectra were recorded using a spectrometer operating in a field of 14.4 T (600 MHz) for ^1^H, (151 MHz) for ^13^C. Chemical shifts are given in parts per million relative to the internal standard tetramethylsilane (^1^H and ^13^C) or relative to the residual peak of the solvents. The 151 MHz ^13^C spectra were acquired under proton decoupling conditions with a 36,000 Hz spectral width, 5.5 μs (60° tip angle) pulse width, 1 s acquisition time, and a 5 s delay time. The ^13^C signals were assigned by distortionless enhancement by polarization transfer spectra (DEPT 1.5).

Dynamic NMR. Temperature calibrations were performed using a digital thermometer and a Cu/Ni thermocouple in an NMR tube filled with 1,1,2,2-tetrachloroethane. The experimental conditions were kept as equal as possible with all subsequent work. The uncertainty in temperature measurements can be estimated as ±1 °C. Line shape simulations were performed using a PC version of the QCPE DNMR6 program. Electronic superimposition of the original spectrum and the simulated one enabled the determination of the most reliable rate constant. The reported values are relative to the exchange from the less populated to the more populated conformation. The rate constants afforded the free energy of activation ΔG^#^ at each temperature by applying the Eyring equation. Within the experimental uncertainty due to the exact temperature determination, the activation energies were found to be invariant, thus implying a small activation entropy ΔS^#^.

### 3.7. DFT Calculations

DFT calculations: Ground state optimizations and transition states were obtained by DFT calculations performed by the Gaussian16 software suite [30] using standard parameters. Full optimization and frequency analysis for ground and transition states employed the B3LYP and the 6-31G(d) basis set. The IEFPCM approach was used to account for the solvent contribution. The analysis of the vibrational frequencies showed the absence of imaginary frequencies for the ground states and the presence of one imaginary frequency for each transition state. Visual inspection of the corresponding normal mode validated the identification of the transition states.

### 3.8. ECD Measurements

The ECD spectra of compounds **1b/2b** were acquired in the 190–400 nm region using a JASCO J-810 spectropolarimeter in far-UV HPLC-grade acetonitrile solution. Concentration was about 1 × 10^−4^ M, tuned by dilution to have a maximum absorbance between 0.8 and 1 with a cell path of 0.2 cm. The spectra were obtained by the average of 6 scans at 50 nm∙min^−1^ scan rate.

### 3.9. Synthesis and Characterization

#### 3.9.1. Synthesis of 1,3-bis((3-(2-methylnaphthalen-1-yl)prop-2-yn-1-yl)oxy)benzene **5**

In a 50 mL round bottom flask equipped with a magnetic stirring bar, resorcinol (144.8 mg, 1.3 mmol, 1 equiv.) was dissolved in DMF (6 mL) at room temperature. Then K_2_CO_3_ (545 mg, 3.95 mmol, 3 equiv.) was added, and after 15 min 1-(3-bromoprop-1-yn-1-yl)-2-methylnaphthalene (750 mg, 2.89 mmol, 2.2 equiv.) was added to the mixture. The reaction was monitored by TLC until the disappearance of the starting material, then diluted with Et_2_O and washed with NaHSO_4_ (×2) and brine (×2). The organic extract was dried over Na_2_SO_4_ and concentrated under reduced pressure. The residue was purified by chromatography on SiO_2_ (25–40 μm), eluting with a 98/2 (*v*/*v*) n-hexane/AcOEt mixture to obtain 589.0 mg (97% yield) of 1,3-bis((3-(2-methylnaphthalen-1-yl)prop-2-yn-1-yl)oxy)benzene 5.

5: pale yellow solid; 97% yield; ^1^H NMR (400 MHz) (CDCl_3_) δ 8.25 (d, *J* = 8.4 Hz, 2 H), 7.79 (d, *J* = 8.1 Hz, 2 H), 7.52 (t, *J* = 7.3 Hz, 2 H), 7.42 (t, *J* = 7.5 Hz, 2 H), 7.36 – 7.28 (m, 3 H), 6.97 (m, 1 H), 6.84 (dd, *J*_1_ = 8.2 Hz, *J*_2_ = 2.2 Hz, 2 H), 5.17 8s, 4 H), 2.60 8s, 6 H); ^13^C{^1^H} NMR (101 MHz) (CDCl_3_) δ 159.0 (q), 139.8 (q), 133.7 (q), 131.4 (q), 130.0 (CH), 128.5 (CH), 128.0 (CH), 127.9 (CH), 126.9 (CH), 125.8 (CH), 125.5 (CH), 118.3 (q), 108.3 (CH), 103.2 (CH), 57.0 (CH_2_), 21.3 (CH_3_).

#### 3.9.2. Synthesis of 4,6-bis(2-bromophenyl)-2,9-dihydropyrano [4,3-g]chromene **1a** and 4,10-bis(2-bromophenyl)-2H,8H-pyrano [2,3-f]chromene **2a**

Compounds **1a** and **2a** were prepared as reported [24]. Linear and angulated **1a**/**2a** were separated by semipreparative HPLC on silica. Column: Silica Adamas (250 × 10 mm ID), eluent Hex/DCM 50/50, flow: 1 mL/min, detector: UV 254 nm.

1a (mixture of stereoisomers): white solid; m.p. = 159–161 °C; IR (neat): 2925, 2837, 1676, 1576, 1488, 1427 cm^−1^; ^1^H-NMR (400.13 MHz, CDCl_3_, +25 °C): δ = 7.45 (m, 2 H), 7.19–7.16 (m, 3 H), 7.06 (dt, *J*_1_ = 7.6 Hz, *J*_2_ = 1.8 Hz, 3 H), 6.42 (s, 1 H), 5.80–5.74 (m, 1 H), 5.55 (s, 2 H), 4.95–4.88 (m, 4 H); ^13^C{^1^H} NMR (101 MHz, DMSO-d6, +80 °C): δ = 155.4, 138.7, 136.0, 132.6, 131.2, 129.1, 127.2, 123.7, 123.2, 118.5, 116.4, 104.0, 65.7; HRMS: *m*/*z* [M + H]^+^ calcd for C_24_H_17_Br_2_O_2_: 496.9569; found: 496.9565.

2a (mixture of stereoisomers): yellow oil; IR (neat): 2930, 2835, 1676, 1575, 1490, 1427 cm^−1^; ^1^H NMR (400 MHz) (CDCl_3_): δ = 7.55–7.47 (m, 2 H), 7.27–6.98 (m, 6 H), 6.36 (s, 2 H), 5.65 (bs, 1 H), 5.39 (bs, 1 H), 4.76–4.61 (m, 2 H), 4.42–4.33 (m, 1 H), 4.22–4.14 (m, 1 H); ^13^C{^1^H} NMR (101 MHz) (CDCl_3_): δ = 155.7, 150.6, 142.6, 139.4, 136.5, 135.3, 133.0, 132.0, 131.4, 130.0, 129.3, 128.3, 127.5, 127.0, 126.3, 123.8, 122.6, 122.2, 118.9, 117.8, 112.4, 109.3, 65.0, 64.9; HRMS: *m*/*z* [M + H]^+^ calcd for C_24_H_17_Br_2_O_2_: 496.9569; found: 496.9565.

#### 3.9.3. Synthesis of 4,6-bis(2-methylnaphthalen-1-yl)-2H,8H-pyrano [3,2-g]chromene **1b** and 4,10-bis(2-methylnaphthalen-1-yl)-2H,8H-pyrano [2,3-f]chromene **2b**

In a 50 mL Carousel Tube Reactor, (Radely Discovery Technology) containing a magnetic stirring bar 1,3-bis((3-(2-methylnaphthalen-1-yl)prop-2-yn-1-yl)oxy)benzene (121 mg, 0.26 mmol, 1 equiv.) was dissolved in CH_2_Cl_2_ (2 mL) at room temperature. Then [tris(2,4-di-tert-butyl-phenyl)phosphite]gold(I) chloride (9.1 mg, 0.01 mmol, 0.04 equiv.) was added, followed by AgSbF6 (3.6 mg, 0.01 mmol, 0.04 equiv.). The mixture was allowed to stir for an hour, and then CH_2_Cl_2_ was evaporated under reduced pressure. The residue was purified by chromatography on SiO_2_ (25–40 μm), eluting with a 97/3 (*v/v*) n-hexane/AcOEt mixture to obtain a mixture of 4,6-bis(2-methylnaphthalen-1-yl)-2*H*,8*H*-pyrano [3,2-g]chromene 1b and 4,10-bis(2-methylnaphthalen-1-yl)-2*H*,8*H*-pyrano [2,3-f]chromene **2b** in a ratio of 87/13. For product characterization, see the following sections.

*Syn/Anti* stereoisomers were separated by semipreparative HPLC on (S,S) Whelk-O_2_ 10 micron (250 × 10 mm L × ID) by using hexane/dichloromethane 95/5 + 0.1% ethanol at a flow rate of 4.0 mL/min. Detector UV 254 nm. In the analytical version, the geometry of the column was 150 × 4.6 mm L × ID (chromatographic trace in Appendix A (black trace). *Anti* and *syn* stereoisomers were assigned based on CD signals (see the red trace in the figure). In addition, changing chiral stationary phase (R,R)-Whlek-O1 5 micron (150 × 4.6 mm L × ID) and using hexane/dichloromethane 95/5 as eluent, the separation of both enantiomers of *anti* 1b was obtained (see Appendix A, at flow of1.0 mL/min, detectors: UV 254 nm, CD 280 nm). After separation, ^1^H-NMR spectra of two stereoisomers were acquired.

1b *syn*: ^1^H NMR (600 MHz) (CDCl_3_): δ 7.51 (d, *J* = 5.4 Hz, 2H), 7.44 (d, *J* = 5.5 Hz, 2H), 7.37 (d, *J* = 5.6 Hz, 2H), 7.16–7.13 (m, 2H), 7.06 (d, *J* = 5.6 Hz, 2H), 6.97–6.94 (m, 2H), 6.48 (s, 1H), 5.48 (t, *J* = 2.4 Hz, 2H), 5.16 (s, 1H), 5.00–4.94 (m, 4H), 2.13 (s, 6H); ^13^C{^1^H} NMR (150 MHz) (CDCl3): d (ppm) 155.4, 133.8, 133.0, 132.8, 131.9, 131.4, 127.7, 127.3, 127.1, 125.5, 125.1, 124.5, 123.0, 118.5, 117.1, 103.6, 65.9, 19.9.

1b *anti*: ^1^H NMR (600 MHz) (CDCl_3_): δ 7.68 (d, *J* = 5.3 z, 2H), 7.63 (d, *J* = 5.5 Hz, 2H), 7.46 (d, *J* = 5.6 Hz, 2H), 7.35–7.27 (m, 4H), 6.86 (d, *J* = 5.6 Hz, 2H), 6.48 (s, 1H), 5.48 (t, *J* = 2.4 Hz, 2H), 5.14 (s, 1H), 5.03–4.94 (m, 4H), 1.79 (s, 6H); ^13^C{^1^H} (150 MHz) (CDCl_3_): δ (ppm) 155.5, 133.7, 133.6, 132.6, 132.0, 131.7, 128.0, 127.6, 127.0, 125.6, 125.5, 124.5, 123.0, 118.5, 117.1, 103.6, 65.9, 19.6.

2b *syn*: ^1^H NMR (600 MHz) (C2D2Cl4): d (ppm) 7.81–7.76 (m, 2 H), 7.72 (d, *J* = 5.3 Hz, 1 H), 7.67 (d, *J* = 5.5 Hz, 2 H), 7.55 (d, *J* = 5.6 Hz, 1 H), 7.38–7.35 (m, 2 H), 7.33–7.29 (m, 3 H), 7.26–7.24 (m, 2 H), 6.26 (d, *J* = 5.6 Hz, 1 H), 6.12 (d, *J* = 5.6 Hz, 1 H), 5.66 (t, *J* = 2.7 Hz, 1 H), 5.26 (t, *J* = 2.5 Hz, 1 H), 4.87–4.81 (m, 2 H), 3.94 (dd, J1 = 9.8 Hz, J2 = 2.6 Hz, 1 H), 3.73 (dd, J1 = 9.8 Hz, J2 = 2.5 Hz, 1 H), 2.35 (s, 3 H), 2.21 (s, 3 H); ^13^C{^1^H} NMR (150 MHz) (C2D2Cl4): d (ppm) 155.6, 151.2, 137.4, 134.1, 134.0, 133.8, 132.7, 132.6, 132.5, 131.9, 131.8, 128.8, 128.6, 128.0, 127.9, 127.6, 126.6, 126.2, 126.0, 125.9, 125.6, 125.1, 124.6, 122.3, 119.7, 118.7, 113.4, 109.1, 65.3, 64.7, 20.9, 20.4.

2b *anti*: ^1^H NMR (600 MHz) (C_2_D_2_Cl_4_): δ 7.78–7.76 (m, 2 H), 7.73 (d, *J* = 5.0 Hz, 1 H), 7.67 (d, *J* = 5.6 Hz, 2 H), 7.62 (d, *J* = 5.5 Hz, 1 H), 7.36–7.28 (m, 6 H), 7.26–7.24 (m, 2 H), 6.26 (d, *J* = 5.6 Hz, 1 H), 6.12 (d, *J* = 5.6 Hz, 1 H), 5.66 (t, *J* = 2.7 Hz, 1 H), 5.25 (, t, *J* = 2.5 Hz, 1 H), 5.25 (s, 1 H), 4.87–4.81 (m, 2 H), 3.96 (dd, *J*_1_ = 9.8 Hz, *J*_2_ = 2.6 Hz, 1 H), 3.66 (dd, *J*_1_ = 9.8 Hz, *J*_2_ = 2.4 Hz, 1 H), 2.39 (s, 3 H), 2.16 (s, 3 H); ^13^C{^1^H} NMR (150 MHz) (C_2_D_2_Cl_4_): δ (ppm) 155.6, 151.2, 137.4, 134.2, 134.0, 133.8, 132.7, 132.63, 132.60, 132.5, 131.9, 131.8, 128.8, 128.7, 128.0, 127.9, 127.6, 126.6, 126.2, 126.0, 125.9, 125.8, 125.6, 125.1, 124.6, 122.3, 119.7, 118.7, 113.4, 109.1, 65.3, 64.7, 21.0, 20.4.

### 3.10. ESI-HRMS 

High-resolution Spectra were recorded on an Exactive Orbitrap Spectrometer (Thermo Scientific) with ESI source. Samples were dissolved in acetonitrile (c: 10^−4^M).

## Data Availability

Not applicable.

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
