# Peer review of "Diaryl-Pyrano-Chromenes Atropisomers: Stereodynamics and Conformational Studies"

_molecules, 2023, doi:10.3390/molecules28134915_

Round 1

Reviewer 1 Report

The manuscript submitted by A. Ciogli et al. reports the synthesis of diaryls-pyrano-chromenes with high racemization barriers that could be resolved into individual diasteromers. The thermodynamics and kinetics of racemization were studied in detail by VT-NMR and dynamic HPLC, and the conclusions were confirmed by extensive DFT and TD-DFT calculations.

An enormous amount of experimental and theoretical work has gone into this study, which fully justifies the possibility of accepting this work after minor technical corrections:

1.       According to the author guidelines “In the text, reference numbers should be placed in square brackets [ ], and placed before the punctuation; for example [1], [1–3] or [1,3].” In the submitted manuscript all reference numbers are placed after the punctuation.

2.       I think that the term “diaryls” in the title looks stange, “diaryl” should be more appropriate.

3.       I feel that I don’t fully understand the end of the sentence starting at line 93 “The free energy term as is, or after frequency cut-off at 100cm-1 (table S6 in SI)[26] and also including empirical 94 dispersion as B3LYP-D3 (table S7 in SI), was persistently off with the experimental data.” Could you please rephrase this “was persistently off with the experimental data”.

4.       If DFT calculations were performed to identify the relative energies of different diastereomers and corresponding transition states, it might be appropriate to add the energy vs. dihedral angle profiles.

5.       Can you assume that the mode of interaction between 1a and R-TFAE can lead to spectral discrimination between syn and anti diastereomers?

6.       Have you attempted to use GIAO calculations to predict the appearance of NMR spectra of different diastereomers and have additional checks on your conclusions? I don't insist on doing such calculations, I'm just interested if it might help.

7.       This work can be a good guideline on how to separate and characterize compounds with restricted rotation. This, for better educational presentation, the equations used to calculate barriers from VT-NMR could be helpful.

8.       Section 2.7 - the phrase "while they were multiplied by a factor" in the caption is misleading, it should probably read "while your rotations were multiplied...". The same correction should be made to the corresponding captions in the Supporting Information.

corrections are indicated in the reviewer report

Author Response

Dear Reviewer,

we thank you for appreciating our work and for having evaluated it suitable for publication in Molecules. All your comments will contribute to a better quality of the manuscript. Changes in the revised version were made in red. In addition, point by point answers are listed below.

The manuscript submitted by A. Ciogli et al. reports the synthesis of diaryls-pyrano-chromenes with high racemization barriers that could be resolved into individual diasteromers. The thermodynamics and kinetics of racemization were studied in detail by VT-NMR and dynamic HPLC, and the conclusions were confirmed by extensive DFT and TD-DFT calculations. An enormous amount of experimental and theoretical work has gone into this study, which fully justifies the possibility of accepting this work after minor technical corrections:

  1. According to the author guidelines “In the text, reference numbers should be placed in square brackets [ ], and placed beforethe punctuation; for example [1], [1–3] or [1,3].” In the submitted manuscript all reference numbers are placed after the punctuation. We modified the position of reference numbers.
  2. I think that the term “diaryls” in the title looks stange, “diaryl” should be more appropriate. We agree with your comment. Title was changed.
  3. I feel that I don’t fully understand the end of the sentence starting at line 93 “The free energy term as is, or after frequency cut-off at 100cm-1 (table S6 in SI)[26] and also including empirical 94 dispersion as B3LYP-D3 (table S7 in SI), was persistently off with the experimental data.” Could you please rephrase this “was persistently off with the experimental data”. We rephrase this sentence with: “did not fit with the experimental data”
  4. If DFT calculations were performed to identify the relative energies of different diastereomers and corresponding transition states, it might be appropriate to add the energy vs. dihedral angle profiles. We report the values of dihedral angle in all figures with GSs and TSs.
  5. Can you assume that the mode of interaction between 1a and R-TFAE can lead to spectral discrimination between syn and anti diastereomers? We cannot assume the mode of interaction but we can discriminate a symmetric 1H NMR signal in the syn conformation to the asymmetric 1H NMR signal in the anti conformation. We modified the text (now lines 160-166) aiming to clarify this approach.
  6. Have you attempted to use GIAO calculations to predict the appearance of NMR spectra of different diastereomers and have additional checks on your conclusions? I don't insist on doing such calculations, I'm just interested if it might help. We have tried to use GIAO calculations for a compound but they have not given accurate results. So, we decided not to show them.
  7. This work can be a good guideline on how to separate and characterize compounds with restricted rotation. This, for better educational presentation, the equations used to calculate barriers from VT-NMR could be helpful. We agree with your comment. We added “Eyring equation” in the sentence:” By line shape simulations of the spectra at different temperatures, and using Eyring equation, a DG¹ value…”
  8. Section 2.7 - the phrase "while they were multiplied by a factor" in the caption is misleading, it should probably read "while your rotations were multiplied...". The same correction should be made to the corresponding captions in the Supporting Information. Done!

Reviewer 2 Report

The manuscript "Diaryls-pyrano-chromenes atropisomers: stereodynamics and conformational studies" by Mancinelli et coll. is a remarkable case study on atropisomers, with determination of rotational barriers, chromatographic separation and absolute configuration assignment. Authors are world recognized expert in the field, they used to study stereodynamic chemistry by VT-NMR and dynamic chiral chromatography. This article is interesting, the science is robust, the conclusions are well supported by the experimental data. In the references part, some articles should be added : general reviews on heterocyclic atropisomers (Clayden, Elguero), references on steric scale for atropisomers could improve the discussion on page 9, and references on chiroptical detection could help the reader. At least, one reference on atropisomeric molecular triads should be added: doi.org/10.1002/chir.10127

Racemization is often used instead of diastereomerization, in particlar in the legends of the SI. The use of theses terms should be checked. “rotational barrier” could also be used.

Other minor remarks:

        page 1, line 28 : helicenes are not atropisomers

        page 9, line 240 : “Chiralpak IB-N5”

        page 13, line 356 : “racemic Whelk O1” is not clear

        Joule should be used for energy in the international system of units.

Author Response

Dear Reviewer,

we thank you for the positive evaluation of our work. Additional references were introduced as your suggestion. The list of selected references is reported below following a “point-by-point answer” format. All changes in the revised version were made in red.

Thank you again for your consideration.

The manuscript "Diaryls-pyrano-chromenes atropisomers: stereodynamics and conformational studies" by Mancinelli et coll. is a remarkable case study on atropisomers, with determination of rotational barriers, chromatographic separation and absolute configuration assignment. Authors are world recognized expert in the field, they used to study stereodynamic chemistry by VT-NMR and dynamic chiral chromatography. This article is interesting, the science is robust, the conclusions are well supported by the experimental data. In the references part, some articles should be added : general reviews on heterocyclic atropisomers (Clayden, Elguero), references on steric scale for atropisomers could improve the discussion on page 9, and references on chiroptical detection could help the reader. At least, one reference on atropisomeric molecular triads should be added: doi.org/10.1002/chir.10127

In accordance with your suggestions, we include additional references:

11. Dynamic Kinetic Resolution and Dynamic Kinetic Asymmetric Transformation of Atropisomers, in Science of Synthesis: Dynamic Kinetic Resolution (DKR) and Dynamic Kinetic Asymmetric Transformations (DYKAT), Editor Bäckvall, J.-E., Thieme Publishing Group, Stuttgart, Germany, 2023, Volume 1, pp. 441–483.

12. Atropisomerism and Axial Chirality in Heteroaromatic Compounds in Advances in Heterocyclic Chemistry, Editor Katritzky, A., Elsevier B.V., Amsterdam, The Netherlands, 2012; Volume 105, pp. 1–188.

13. GawroÅ„ski, J.; Kacprzak, K. Architecture and function of atropisomeric molecular triads, Chirality 2002, 14, 689–702.

37. Belot, V.; Farran, D.; Jean, M.; Albalat, M.; Vanthuyne, N.; Roussel C. Steric Scale of Common Substituents from Rotational Barriers of N-(o-Substituted aryl)thiazoline-2-thione Atropisomers, Org. Chem. 2017, 82, 19, 10188–10200.

38. Chen, Y.-H.; Li, H.-H.; Zhang, X.; Xiang, S.-H., Li, S.; Tan B. Organocatalytic Enantioselective Synthesis of Atropisomeric Aryl-p-Quinones: Platform Molecules for Diversity-Oriented Synthesis of Biaryldiols Chem. Int. Ed. 2020, 59, 11374–11378.

39. Sweet, J. S.; Rajkumar, S.; Dingwall, P.; Knipe, C. Atroposelective Synthesis, Structure and Properties of a Novel Class of Axially Chiral N-Aryl Quinolinium Salt Eur. J. Org. Chem. 2021, 3980–3985.

Racemization is often used instead of diastereomerization, in particlar in the legends of the SI. The use of theses terms should be checked. “rotational barrier” could also be used. We carefully revised this aspect and appropriate changes were done (lines xx and xx in Manuscript, several in SI).

Other minor remarks:

–        page 1, line 28 : helicenes are not atropisomers. Sorry for mistakes, we delete it immediately.

–        page 9, line 240 : “Chiralpak IB-N5”. Done.

–        page 13, line 356: “racemic Whelk O1” is not clear. It’s a copy/paste error, as reported in page S16, we used the (S,S) Whelk-O2 for syn/anti separation.

–        Joule should be used for energy in the international system of units. All kcal/mol values were now also in kJ/mol.